# Yawn contagion in humans and bonobos: emotional affinity matters more than species

Elisabetta Palagi[1,2], Ivan Norscia[1] and Elisa Demuru[1,3]

[1] Natural History Museum, University of Pisa, Pisa, Italy
[2] Institute of Cognitive Sciences and Technologies, Unit of Cognitive Primatology & Primate Center, CNR, Rome, Italy
[3] Bioscience Department, University of Parma, Parma, Italy

## ABSTRACT

In humans and apes, yawn contagion echoes emotional contagion, the basal layer of empathy. Hence, yawn contagion is a unique tool to compare empathy across species. If humans are the most empathic animal species, they should show the highest empathic response also at the level of emotional contagion. We gathered data on yawn contagion in humans (*Homo sapiens*) and bonobos (*Pan paniscus*) by applying the same observational paradigm and identical operational definitions. We selected a naturalistic approach because experimental management practices can produce different psychological and behavioural biases in the two species, and differential attention to artificial stimuli. Within species, yawn contagion was highest between strongly bonded subjects. Between species, sensitivity to others' yawns was higher in humans than in bonobos when involving kin and friends but was similar when considering weakly-bonded subjects. Thus, emotional contagion is not always highest in humans. The cognitive components concur in empowering emotional affinity between individuals. Yet, when they are not in play, humans climb down from the empathic podium to return to the "understory", which our species shares with apes.

Corresponding author
Elisabetta Palagi,
elisabetta.palagi@unipi.it

## INTRODUCTION

Most behavioral and cognitive studies have used the most sophisticated human abilities (e.g., theory of mind and language) as the main ground to make comparative arguments. Yet, this top-down approach has contributed to drawing the line between human and other animals' skills. Indeed, selecting the emotional and cognitive "pinnacles of mental evolution" the human brain (*sensu de Waal & Ferrari, 2010*, p. 201) can shed light on unique human traits but not on the common basic mechanisms underlying the more complex ones (*de Waal & Ferrari, 2010*; *Hecht et al., 2012*). While the hominin fossil record cannot give us any clues on the social and emotional abilities that may have paved the way for the emergence of human empathy, studying synchronous behaviours and body resonance (e.g., laughing, rapid facial mimicry, yawning; *Provine, 2012*; *Davila-Ross,*

*Menzler & Zimmermann, 2008*; *Mancini, Ferrari & Palagi, 2013a*) in our closest living relatives may provide valuable information about how empathy evolved.

Empathy is a multilayered phenomenon, whose shared basal forms represent the pre-existing framework in which more complex emotional and cognitive processes are nestled (*Preston & de Waal, 2002*; *de Waal, 2008*). Perceiving and sharing others' emotions, the so called "affective empathy", is a phylogenetically old capacity. Affective empathy is an automatic process and may come about through emotional contagion (*de Waal, 2008*). At a higher layer of the empathic sphere, we find "cognitive empathy" which requires self-other differentiation, perspective-taking and mental state attribution, basic capacities to infer others' emotional states. Affective and cognitive empathy dissociate in humans with the former preceding the latter both ontogenetically and phylogenetically (*Decety, 2010*). Humans show the most complex form of empathy and, therefore, are commonly deemed as the most empathic species (*Preston & de Waal, 2002*; *Decety & Jackson, 2004*; *Decety & Cowell, 2014*).

However, no quantitative, direct inter-specific comparison has ever been made to actually demonstrate that this assumption is true at each floor of the empathic building. Recent findings endorse that other animals show affective empathy (*Palagi et al., 2009*; *de Waal, 2012a*; *Panksepp & Panksepp, 2013*). In this view, cross-species research is needed to explore empathy capacities as a bottom-up, emotional and developmental process of the brain (*de Waal, 2012b*). According to several neurobiological (*Cooper et al., 2012*; *Haker et al., 2013*), psychological (*Lehmann, 1979*; *Platek et al., 2003*) and ethological findings (*Palagi et al., 2009*; *Campbell & de Waal, 2011*; *Campbell & de Waal, 2014*; *Romero, Konno & Hasegawa, 2013*) yawn contagion is an empathy-related phenomenon. Specifically, yawn contagion is a form of emotional contagion, which represents the most basal layer of the empathic domain (e.g., see the "Russian Doll Model" in *de Waal, 2008*; *Preston & de Waal, 2002*; *Hatfield, Rapson & Le, 2009*). Several good reasons make the yawn and its contagion excellent phenomena to explore the evolutionary roots of empathy.

Due to its evolutionary antiquity, a yawn can be easily detected and quantified, appearing morphologically identical across many different vertebrate taxa and, consequently, withdrawing the risk of any subjective interpretation (*Provine, 2005*). Therefore, the plesiomorphic nature of yawning enables cross-species research (*Baenninger, 1987*; *Baenninger, 1997*; *Deputte, 1994*; *Guggisberg et al., 2010*). Yawning is automatic, unstoppable and easily recognisable also when people try to hide or inhibit it due to cultural constraints (*Lehmann, 1979*). These features make this behaviour an honest signal and, therefore, highly reliable (*Provine, 1986*).

Yawn contagion is obvious in human beings—about 50 percent respond to video stimuli of yawning faces (*Provine, 1986*)—and seems to be based on a perception-action mechanism (*Preston & de Waal, 2002*; *de Waal, 2012c*), which consists in the involuntary re-enactment of an observed facial expression and creates shared representations. Neurophysiological evidence of this coupling has derived from the discovery of mirror neurons (*di Pellegrino et al., 1992*), recently associated with yawn contagion (*Cooper et al., 2012*; *Haker et al., 2013*). Therefore, yawn contagion provides a "low-tech" but significant

evidence of mirror-like phenomena. Mirror neurons, firstly described in the pre-frontal cortex of the rhesus macaque, fires when the subject either performs a motor action or observes the same action performed by another subject (*Gallese et al., 1996*; *Ferrari et al., 2003*). A mirror system, homologous to that of macaques, has been later discovered in humans (*Rizzolatti & Craighero, 2004*). The mirror mechanism (or motor mimicry), not requiring any kind of conscious awareness, creates an emotional bridge and, thanks to it, two individuals can synchronize their affective states (*Hatfield, Rapson & Le, 2009*). This emotional bridge, in turn, fosters mirroring and triggers positive feedback which constitutes the core of social affinity. Such feedback is more easily activated when subjects sharing a strong empathic bonding are involved. While a positive attachment enhances body and emotional resonance (e.g., emotional reactions, unconscious mimicry, self-other overlap and shared representations), a negative attachment or a prejudice can inhibit or suppress it (*Xu et al., 2009*; *Avenanti, Sirigu & Aglioti, 2010*). In humans, when the attachment is profoundly negative (as in case of defectors, enemies, competitors) empathy can even turn into *Schadenfreude* that is feeling pleasure from the sufferings of others (*de Waal, 2008*; *Decety & Cowell, 2014*). This emotional circuitry, in its positive and negative form, has been found not only in humans (*Singer et al., 2006*; *Pfeifer et al., 2008*) but also in monkeys (*Masserman, Wechkin & Terris, 1964*; *Palagi et al., 2009*; *Mancini, Ferrari & Palagi, 2013a*; *Mancini, Ferrari & Palagi, 2013b*; *Ferrari, Bonini & Fogassi, 2009*; *Paukner et al., 2009*) and apes (*Anderson, Myowa-Yamakoshi & Matsuzawa, 2004*; *Davila-Ross, Menzler & Zimmermann, 2008*; *Campbell & de Waal, 2011*; *Campbell & de Waal, 2014*).

In humans (*Homo sapiens*) and one of its closest living phylogenetic relatives, the bonobo (*Pan paniscus*), yawn contagion is present and significantly affected by the emotional closeness linking the responder to the first yawner (the trigger). In the two species, the yawn response is most likely when the yawning stimulus comes from kin or friends (*Norscia & Palagi, 2011*; *Demuru & Palagi, 2012*). Since both in humans and bonobos yawn contagion is a socially modulated phenomenon, these species are good models to test some hypotheses on the evolutionary origins of the linkage between yawn contagion and empathy. The opportunity provided by yawn contagion to apply the same unit of measurement and identical operational definitions permits to directly compare human empathy with that of other species. Indeed, most of the studies on empathic capacities in *Homo sapiens* have been carried out through questionnaires used to measure self-reported scores of empathy. This approach bears the risk of overestimating the human empathic potential and leads to the unfeasibility of comparing it to that of other species (*Lawrence et al., 2004*).

As a whole, the possibility of employing a highly straightforward behaviour, such as yawning, to explore a highly complex phenomenon, such as empathy, represents a unique opportunity to investigate and compare empathy in human and non-human species (*de Waal, 2012b*).

Theoretically, humans could appear as the most empathic species not because they experience greater emotional contagion (affective empathy) but because they can better understand others' perspective and simulate others' emotional experiences (cognitive empathy). Indeed, empathy is a construct comprising dissociable components interacting

and operating in parallel (*Decety & Cowell, 2014*). Cognitive empathy can be impaired in autistic subjects (*Dziobek et al., 2008*; *Usui et al., 2013*) whereas emotional empathy can be dramatically impaired in narcissistic individuals (*Ritter et al., 2011*), criminal psychopaths (*Woodworth & Porter, 2002*), rapists (*Englander, 2007*), and child molesters (*Marshall, Hamilton & Fernandez, 2001*). Yet, both emotional and cognitive empathy contribute to physiological human empathy (*Cox et al., 2011*; *Decety & Cowell, 2014*). Therefore, if humans are actually the most empathic species, they should possess the highest empathic response starting with emotional contagion, the ground level of affective empathy.

We tested this hypothesis by comparing our species with bonobos, which diverged from the human line about 5–7 mya (*Fleagle, 2013*). We expect that the frequency of yawn response is always higher in humans than in bonobos, whatever the relationship quality linking the responder to the trigger. Since the time elapsing from the yawn stimulus to the response can be reduced when the social bond is strong (*Norscia & Palagi, 2011*), we also expect that the emotional affinity linking the subjects affects, more in humans than in bonobos, the promptness of the yawn response.

## METHODS

### Ethics statement

This study was purely observational, with no manipulation whatsoever. For humans, information was entered in an anonymous form. Thus, the ethics committee of the University of Pisa waived the need for a permit.

### Data collection

We gathered data on yawn contagion in both species in their daily life environment, by applying the same observational paradigm based on identical operational definitions. The naturalistic approach has been selected because it permits collection of reliable data thanks to the absence of invasive, artificial management practices (social isolation, spatial limitation, interaction with novel objects and persons) which can produce different psychological and behavioural biases in the two species, and differential attention to artificial stimuli. For example, previous research on humans has demonstrated that the awareness of being observed can inhibit yawns (*Provine, 2005*) whereas simply reading about yawning is sufficient to trigger yawns (*Provine, 1986*) and possibly inflate yawn response in the experimental subjects (*Bartholomew & Cirulli, 2014*).

*Humans*—Naturalistic observations on humans were performed over a total of 380 h (in 2010, 2011 and 2013) and involved 33 adults, 19 females and 14 males. The study subjects were observed during their everyday activities in their natural social setting (at work, in restaurants, waiting rooms, during social meals, etc.). In public spaces, the authors sat down close to the study subjects and observed. Subjects included people known to the authors, such as friends, family members, coworkers, and students. The study also included individuals that the authors did not know but whose personal information (e.g., country of origin and social bond with other study subjects) was stated by the observed subjects during conversations. Data were typed and stored in mobile phones (e.g., during dinners),

entered in the laptop (when possible, e.g., on the train), or noted down on paper (e.g., in public spaces where this practice could easily go unnoticed). Data collection occurred only when at least five subjects were present to make observations on humans comparable with the observations on bonobos (see below). During each observation bout (spanning 1–4 h), yawns were collected via the all occurrences sampling method (*Altmann, 1974*). Dyads of strangers were excluded. We collected a total of 1,375 yawn events. The observed subjects were not aware of being under investigation and the data collected were entered in an anonymous form with an alphanumerical code uniquely assigned to each subject.

*Bonobos*—Naturalistic observations on bonobos were performed daily on the colonies hosted at the Apenhuel Primate Park (Apeldoorn, The Netherlands) and at the Wilhelma Zoo (Stuttgart, Germany) for a total of 502 and 323 h, respectively (2009/2010). The all occurrences sampling observations involved 16 adults, 12 females and 4 males in different contexts (foraging, resting, feeding both indoor and outdoor). At least five bonobos were always present in the same enclosure. During each observation bout (spanning 2–5 h), yawns were collected via the all occurrences sampling (*Altmann, 1974*). We collected a total of 1,125 yawn events for the Apenheul group and 998 for the Stuttgart group.

Both for humans and bonobos when a subject spontaneously yawned, we recorded (1) time; (2) the encoded identity of the yawner (hereafter, the "trigger") and of each potential responder (hereafter, the "observer"), that is every individual that could visually and/or auditorily perceive the triggering yawn; (3) presence/absence of contagion within 3-min following the last triggering yawn; (4) time latency in the yawn response; (5) trigger's and observer's sex. Before starting systematic data collection, the three observers (the authors) underwent a training period during which they simultaneously followed the same group of subjects. The training procedures were identical for the two species. Each yawning event and the possibility to perceive it by other observed subject were the items we considered for the calculation of the Cohen's kappa. Training ended when the observations produced a Cohen's kappa >0.80. Every 50 h of data collection we checked observational reliability (Cohen's kappa never <0.80).

## Operational definitions

In humans we defined strongly and weakly bonded dyads on the basis of previous literature (*Norscia & Palagi, 2011*). Friends, regular partners, and kin were labeled as strongly bonded dyads. Friends were defined as non related individuals sharing a direct relationship, frequenting each other voluntarily. Regular partners were defined as couples living together from at least one year. Those dyads having a coefficient of relatedness $r \geq 0.25$ were defined as kin (7 dyads, $r = 0.5$; 1 dyad $= 0.25$). Weakly bonded dyads involved acquaintances, that is people who exclusively shared an indirect relationship based on a third external reference, for example work duty (colleagues) or friends in common (friends of friends). The relationship between people was known to the authors. Ambiguous cases were excluded from the dataset (e.g., kin with $r < 0.25$, colleagues frequenting each other outside work).

In bonobos we also categorized strongly and weakly bonded dyads on the basis of previous literature (*Demuru & Palagi, 2012*; *Palagi & Norscia, 2013*). By focal animal

sampling (*Altmann, 1974*—25 h of observation per subject), we were able to record all the contact sitting, grooming, and food sharing sessions performed by each focal animal with any other group member. Each subject was followed every day (each focal lasted 30 min) and at different times to obtain data covering the entire day in balanced proportions as much as possible. Friends and kin were clustered into the strongly bonded category. Friend dyads were categorized using a combined measure of the three behaviors collected during focal observations and calculating the quartile points of dyadic scores for each focal individual. Only dyads with scores falling into the top quartile were considered as friends. Kinship was based on maternal lineages only, as paternity was unknown. Our kin sample consisted in 7 dyads ($r = 0.5$), all showing top frequencies of affinitive contact exchange (strong bond). All the dyads not included in the previous categories (non-kin dyads' scores falling outside the top quartile) were labeled as weakly bonded.

## Statistics

Due to the normality of data (Kruskall-Wallis, $p > 0.05$), we made use of parametric statistics. Independent sample $t$-test was used to compare the frequency and the latency of yawn contagion between the two species.

To explore the effect of the different variables on the frequency of yawn contagion we analyzed data via Linear Mixed Model (LMM). The dependent scale variable was the relative frequency of yawn contagion of each observer; that is, the number of yawns the observer performed after perceiving a given trigger's yawn normalized on the number of occasions (number of a given trigger's yawns perceived by the observer). In the analysis, triggers and observers' identities were entered as random factors (nominal variables). We tested models for each combination involving the variables of interest, spanning from a single-variable model to a model including all the fixed factors (full model). To select the best model, we used the Akaike's Corrected Information Criterion (AICc), a measure for comparing mixed models based on the $-2$ (Restricted) log likelihood. The AICc corrects the Akaike's Information Criterion (AIC) for small sample sizes. As the sample size increases, the AICc converges to AIC. The model with a lower value of AIC was considered to be the best model. To avoid the increase of type II errors, factors were excluded from a model only if this improved the model fit by >2 AICc units. The value of degrees of freedom is given by the effective sample size (N) minus the rank design matrix of fixed effects (X). The denominator degree of freedom is estimated by SPSS via Satterthwaite's approximation.

To check for the inter-specific differences in the frequency of yawn contagion between weakly and strongly bonded dyads we applied the randomization test for two independent samples. This kind of procedure is used to avoid pseudo-replication due to non-independence of data (the same individual is included in more than one dyad; therefore, dyads are not independent data-points). Specifically, randomization tests were employed with a number of 10,000 permutations using resampling procedures (via Resampling Procedures 1.3 package by David C. Howell).

To test for the individual differences in the latency of yawn contagion as a function of relationship quality shared by subjects, we used the paired sample $t$-test.

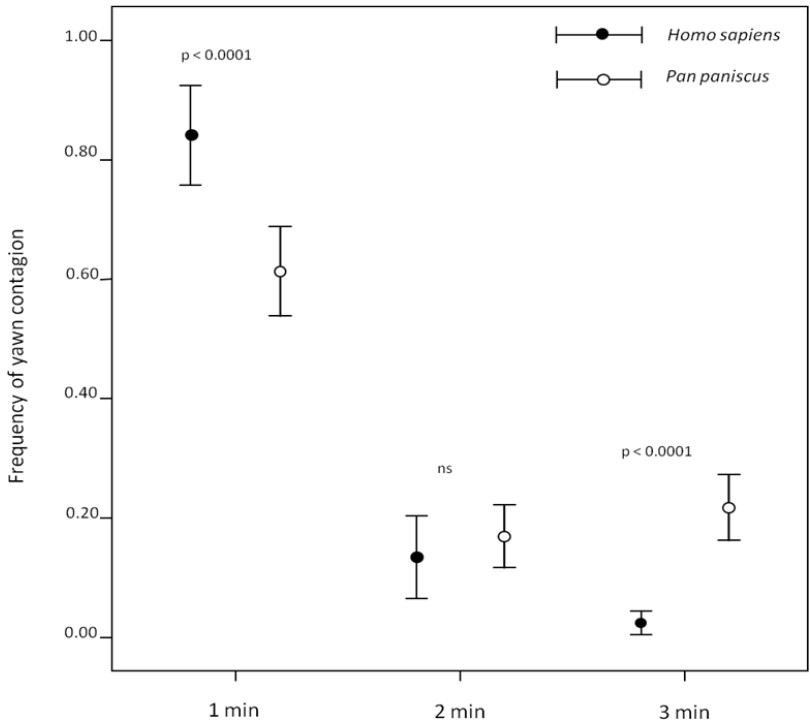

**Figure 1 Timing distribution of yawn contagion in the two species.** The error bars show the mean (±SE) of the individual frequency as a function of the 3-min time window.

## RESULTS

The individual yawn response did not differ between the two species (mean yawn response ±SE: *Homo* = 0.296 ± 0.042; *Pan* = 0.210 ± 0.029; independent sample test: $t = 1.348$; df = 47; $p = 0.184$).

Both humans (*Norscia & Palagi, 2011*) and bonobos (*Demuru & Palagi, 2012*) (Video S1) preferentially responded within the first minute after perceiving a yawn. However, our direct comparison between the two species showed that, within the first minute, humans' responses were more frequent than bonobos' (first minute: independent sample test: $t = 3.854$; df = 37; $p < 0.0001$), who showed a longer tail-effect in their contagion response compared to humans (second minute: $t = -0.737$; df = 37; $p = 0.466$; third minute: $t = -7.555$; df = 37; $p < 0.0001$) (Fig. 1). Since this analysis considered three different minutes in which yawn contagion may occur, only the subjects showing at least three contagion events were included in the test.

We employed Linear Mixed Models (LMM) to verify which variables could explain the variation in the frequency of yawn contagion (dependent variable). Data distribution was normal. Species (*Homo sapiens/Pan paniscus*), trigger and observer's sex, and social bonding (weak/strong) were entered as fixed factors. Trigger and observer's identities were entered as random factors. This analysis was restricted to those dyads in which each contagion event could be univocally assigned to a specific trigger and when the

**Table 1  LMM results.** Best LMM explaining the occurrence of yawn contagion (AICc = −3.312).

|  | Numerator df | Denominator df | F | Significance level |
|---|---|---|---|---|
| Intercept | 1 | 17.771 | 52.037 | 0.000 |
| **Fixed factors** | | | | |
| Social bonding[**] | 1 | 72.536 | 10.610 | 0.002 |
| Species[*] | 1 | 17.380 | 7.621 | 0.013 |
| **Random factors** | **Variance** | **SE** | | |
| Trigger's identity | 0.0333 | 0.017 | | |
| Observer's identity | 0.0038 | 0.004 | | |

Notes.

[*] Significant.

[**] Highly significant.

df, degrees of freedom; SE, standard error.

opportunities were at least 3 per dyad. For these reasons, the LMM dataset included 44 human and 56 bonobo dyads with 375 and 484 contagion events, respectively.

Both species and social bonding remained in the best model (best model AICc = −3.312; worst model$_{intercept only}$ AICc = 6.828) and significantly affected the frequency of yawn contagion (Table 1).

In both humans and bonobos, yawn contagion was higher between individuals sharing strong relationships (randomization test for two independent samples: bonobos, $t = -2.772$, $N_{strong} = 28$ dyads, $N_{weak} = 28$ dyads $p = 0.0061$; humans, $t = -3.646$, $N_{strong} = 28$ dyads, $N_{weak} = 16$ dyads, $p = 0.0007$). At the inter-specific level, the frequency of yawn contagion in humans and bonobos significantly differed only when strongly bonded dyads ($N_{Pan} = 28$; $N_{Homo} = 28$) were involved (randomization test for two independent samples $t = -3.916$, $p = 0.0001$). On the contrary, weakly bonded dyads ($N_{Pan} = 28$; $N_{Homo} = 16$) of the two species did not differ in the frequency of yawn contagion (randomization test for two independent samples $t = -0.799$, $p = 0.455$) (Fig. 2).

In order to understand if the response latency was affected by the relationship quality shared by the trigger and the responder in the two species, we analyzed the minute distribution of the response as a function of weak and strong bonds. In bonobos, the analysis showed that the response latency did not change as a function of the relationship quality (paired sample test: $t_{1 min} = -0.233$, df = 12, $p = 0.820$; $t_{2 min} = 0.196$, df = 12, $p = 0.848$; $t_{3 min} = 0.199$, df = 12, $p = 0.846$; Fig. 3A). On the contrary, in humans, the response latency was sensitive to the relationship quality. The human subjects concentrated their response within the first minute only when the yawn stimulus came from a strongly bonded subject (paired sample test: $t_{1 min} = 4.760$, df = 12, $p = 0.0001$). On the other hand, the human subjects responded more frequently in the second and third minute after perceiving the yawn stimulus from a weakly bonded trigger ($t_{2 min} = -3.774$, df = 12, $p = 0.003$; $t_{3 min} = -2.670$, df = 12, $p = 0.020$; Fig. 3B). All these analyses included only subjects showing at least three contagion response per each bonding category (weak and strong).

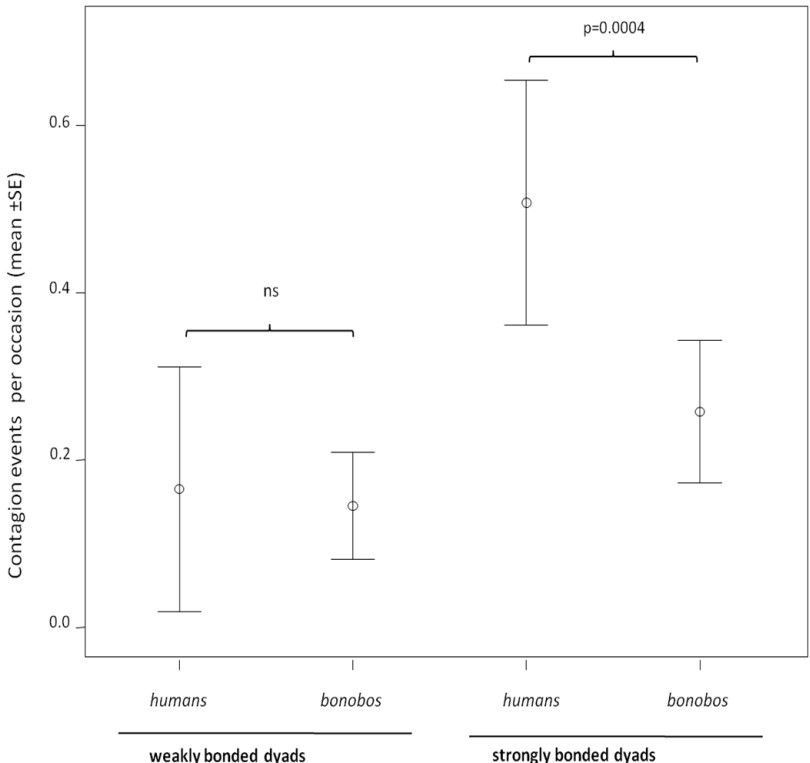

**Figure 2 Yawn contagion and relationship quality in the two species.** Error bars showing the dyadic frequency of yawn contagion (number of responses per yawn stimulus perceived) (mean ± SE) in the two species as a function of the relationship quality linking the subjects involved.

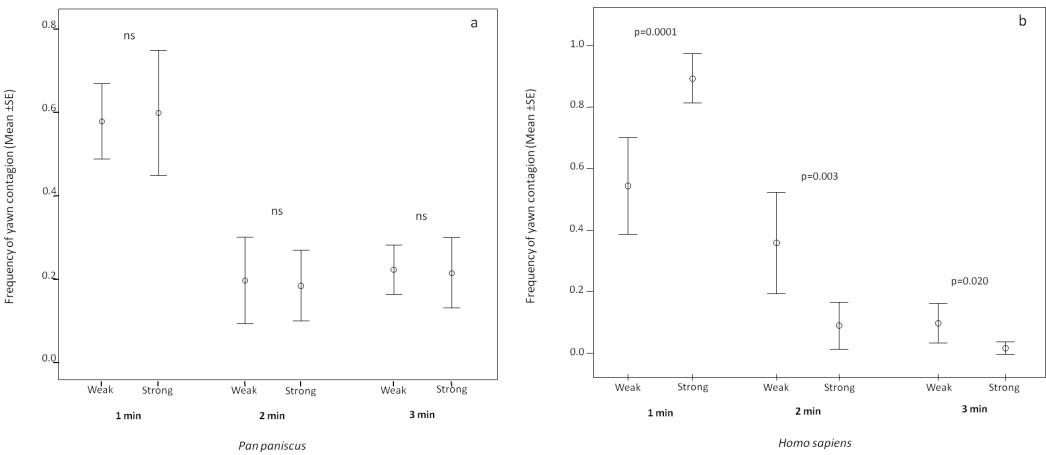

**Figure 3 Latency of yawn contagion as a function of relationship quality.** Error bars (mean ± SE) showing the timing distribution of the dyadic yawn contagion in bonobos (A) and humans (B) as a function of the relationship quality shared by the subjects involved.

## DISCUSSION

The inter-specific analysis of the overall frequency of yawn contagion did not reveal any difference between human and bonobo response susceptibility. The inter-species comparison revealed a higher promptness in humans than in bonobos, who showed a longer tail-effect in their response latency. However, both species concentrated their responses within the first minute after perceiving the stimulus yawn (Fig. 1). As yawn contagion is the expression of a mirror-like phenomenon (*Cooper et al., 2012*; *Haker et al., 2013*), our findings suggest that not only is yawn contagion well-rooted in the biology of bonobos and humans but also that the neural processes underlying and modulating yawn contagion might have been already present in the most recent common ancestor of the two species (5–7 mya, *Fleagle, 2013*). The similar sensitivity to others' yawns shown by the two species is probably due to similar selective pressures. In fact, both species are characterized by obligate gregariousness based on one of the most complex forms of social structure: the fission–fusion system (*Aureli et al., 2008*). Behavioural (*Couzin, 2007*) and emotional synchronization (*Špinka, 2012*) has been playing a pivotal role in favouring and shaping the evolutionary pathways leading to social living (Video S1). Synchronizing with others requires the ability to mirror their motor actions, to experience their emotional states and, consequently, to act in the appropriate manner. In this process familiarity makes the difference. Our intra-specific analysis demonstrated that yawn contagion is a socially modulated phenomenon, with strongly bonded subjects showing a higher susceptibility than weakly bonded subjects, thus confirming previous results (humans, *Norscia & Palagi, 2011*; bonobos, *Demuru & Palagi, 2012*). In this respect, the degree of contagion of around 50% reported by previous studies using videos of strangers' yawns as stimuli (*Provine, 2005*) may be a conservative estimate.

The social modulation of yawn contagion has been proven also in the congeneric species of bonobos, the chimpanzee (*Pan troglodytes*). Chimpanzees are more susceptible to respond to videos showing yawns performed by in-group than by out-group members (*Campbell & de Waal, 2011*). This result is in line with a great body of research on human in-group/out-group bias that affects numerous empathy-driven behavioural and emotional reactions (*Bourgeois & Hess, 2008*; *Xu et al., 2009*; *Avenanti, Sirigu & Aglioti, 2010*). Recently, *Campbell & de Waal (2014)* also found that the rate of chimpanzee response to others' yawns was similar when the stimulus came from an in-group chimpanzee and from a human subject, thus demonstrating that the chimpanzee involuntary empathic response goes beyond the species boundary, as it occurs in humans (*Phillips, 2009*). Yet, no evidence of a social modulation in chimpanzee yawn contagion emerged as a function of the relationship quality shared between the responder and the familiar trigger (*Massen, Vermunt & Sterck, 2012*). Both neuroanatomical (*Rilling et al., 2012*) and behavioral features (*Hare, Wobber & Wrangham, 2012*) can account for the different effect of the relationship quality on chimpanzee and bonobo yawn contagion. Compared to chimpanzees, bonobos seem to show a different emotional and affective sensitivity (*Furuichi, 2011*; *de Waal & Lanting, 1997*; *Hare et al., 2007*) and a milder competitive propensity (*Kano, 1992*; *Tan & Hare, 2013*). Moreover, genetic

findings revealed that a DNA segment regulating the responses to a hormone involved in social bonding (vasopressin) is present in humans and bonobos, but can be absent in chimpanzees (*Hammock & Young, 2005*). However, the different effects of the relationship quality on yawn contagion in the two *Pan* species has to be taken cautiously due to the different methodological approaches used to verify and quantify the phenomenon (video stimulus—*Massen, Vermunt & Sterck, 2012*; living stimulus—*Demuru & Palagi, 2012*).

At the inter-specific level, striking differences emerged in the social modulation of human and bonobo yawn contagion. While strong and weak relationships between individuals had to be assessed according to species-specific criteria (see Methods section), the previous works on bonobos (*Demuru & Palagi, 2012*) and humans (*Norscia & Palagi, 2011*) indicate that the strong-weak categorization produces a similar effect on yawn contagion in either species, with yawn contagion being highest between individuals sharing strong relationships. This result is confirmed in this study. Compared to bonobos, the human susceptibility (Fig. 2) and promptness (Fig. 3) to others' yawns were significantly more potentiated when kin and friends were involved. Humans' responses were more frequent and faster when the trigger and the responder shared a strong emotional bond. On the other hand, susceptibility and promptness incredibly overlapped between the two species when a strong emotional involvement between subjects was lacking, thus indicating that emotional contagion is not always highest in humans. It is worth remarking that these findings would have been impossible to detect if limiting the analysis to the intra-specific level.

What explains the difference in yawn contagion between species is the deepness of the emotional affinity linking the subjects. Compared to bonobos, humans show a different degree of sensitivity at the most basal layer of empathy, but only when they are strongly emotionally involved. Moreover, the latency of yawn contagion is socially modulated in humans (the stronger the emotional involvement, the faster the response) but not in bonobos. Therefore, the positive feedback linking emotional affinity and the mirroring process is more easily and rapidly activated in humans than in bonobos. Such over-activation explains not only the human potentiated yawning response, but also other kinds of unconscious mimicry response, such as happy, pain or angry facial expressions (*Chiao et al., 2008*). Minagawa-Kawai and coworkers (*2009*) investigated the response latency of smiles in mother-infant pairs and found neurobiological support for the "over-activation" hypothesis. The faster response was associated with an increased activation in the regions around the orbitofrontal cortex in mothers while viewing their own infant's smile compared to an unfamiliar infant's smile. In these mothers, specific neuronal regions involved in positive emotional regulation were activated by both viewing familiar and unfamiliar infants but the magnitude of activation was greater when affective attachment was involved. A similar neuro-ethological approach has never been applied to quantify the extent of the neural activation at the basis of the difference in yawn contagion latency as a function of the emotional closeness.

The higher human sensitiveness to yawns emitted by friends and kin can be explained by the peculiarity of the emotional attachment characterizing our species (*Bowlby, 1969*; *Maestripieri, 2003*). Compared to bonobos, the strong relationships established between

humans are probably qualitatively different, because they are built upon more complex and sophisticated emotional foundations linked to cognition, memory, and memories. This interpretation is in line with findings endorsing that affective and cognitive empathy cooperate in modulating the phenomenon of yawn contagion in humans as it occurs for other kinds of unconsciously mimicked behaviours (*Van Baaren et al., 2009*). The idea that also the cognitive component of human empathy explains some features of yawn contagion is supported by the ontogenetic co-emergence of yawn contagion and theory of mind capacities in children (*Anderson & Meno, 2003*; *Helt et al., 2010*; *Millen & Anderson, 2011*) and the correlation of these two phenomena in adults (*Platek et al., 2003*). Therefore, it seems that in humans affective empathy is enhanced by the cognitive components concurring in awakening and nourishing the affective involvement characterizing strong relationships (*Van Baaren et al., 2009*).

The most intriguing and original result emerging from our cross-species analysis is the strong similarity between human and bonobo contagion susceptibility when the stimulus comes from a familiar, but weakly bonded subject. This finding indicates that the foundations of the empathic mechanism are alike in the two species. The absence of a strong emotional involvement seems to bring to light the most basic component of yawn contagion by removing the influence of the higher and more complex layers characterizing the cognitive sphere of human empathy (e.g., perspective taking, mental state attribution, theory of mind). As a whole, our cross-species approach supports the shared and multilayered architecture of animal empathy (*Preston & de Waal, 2002*; *de Waal, 2008*). Therefore, we cannot state that emotional contagion is always highest in humans, but only that they are able to create a unique and extremely intense form of emotional attachment, thanks to the complexity of the neural circuits linking social experiences and cognitive capacities. When such a pervasive attachment is missing, humans climb down from the highest step of the empathic podium to return to the understory layer which our species shares with other great apes.

## ACKNOWLEDGEMENTS

Thanks are due to Frank Rietkerk (Apenheul Primate Park, Apeldoorn, The Netherlands) and Marianne Holtkötter (Wilhelma Zoo, Stuttgart, Germany) and the bonobo keepers for allowing and facilitating this work. We also thank Francesca Coppola and White Palagio for their important clarifying input in discussing results.

### Funding
The authors declare there was no funding for this work.

### Competing Interests
The authors declare there are no competing interests.

## Author Contributions

- Elisabetta Palagi conceived and designed the experiments, performed the experiments, analyzed the data, contributed reagents/materials/analysis tools, wrote the paper, prepared figures and/or tables, reviewed drafts of the paper.
- Ivan Norscia performed the experiments, analyzed the data, contributed reagents/materials/analysis tools, reviewed drafts of the paper.
- Elisa Demuru performed the experiments, analyzed the data, contributed reagents/materials/analysis tools, wrote the paper, prepared figures and/or tables, reviewed drafts of the paper.

## Human Ethics

The following information was supplied relating to ethical approvals (i.e., approving body and any reference numbers):

This study was purely observational, with no manipulation whatsoever. For humans, information was entered in an anonymous form. Thus, the ethics committee of the University of Pisa waived the need for a permit. The observed subjects were not aware of being under investigation and the data collected were entered in an anonymous form with an alphanumerical code uniquely assigned to each subject.

## Animal Ethics

The following information was supplied relating to ethical approvals (i.e., approving body and any reference numbers):

This study was purely observational, with no manipulation whatsoever. Thus, the ethics committee of the University of Pisa waived the need for a permit.

## Supplemental Information

Supplemental information for this article can be found online at http://dx.doi.org/10.7717/peerj.519#supplemental-information.

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
