# Peer review of "Yawn contagion in humans and bonobos: emotional affinity matters more than species"

_PeerJ, doi:10.7717/peerj.519_

## Round 0.1 · original submission · Minor Revisions

In addition to some basic reporting issues affecting the readability of the manuscript, the reviewers have raised some questions about the experimental design. When revising your manuscript, please address the issues regarding the experimental design so that both the methodology and rationales are clear (e.g., recording data only when five subjects are present; exclusion of dyads with less than three contagion events; and any operationalization differences). As a minor issue, one of your references appears to be cited differently in the text and reference section (Davila-Ross et al. 2008 in text but Ross MD et al. in the listed references), so please correct that. It seems that there are no major problems with validity of findings; however, when revising the manuscript, please distinguish speculation and conclusions drawn directly from data you present. It may be necessary to provide additional references for support.

Reviewer 1 ·

Basic reporting

Overall the reporting is appropriate. There are a few language oddities that could perhaps be best explained by the authors not being native English speakers. For example: lines 27-30, 91-94, 98-100, last sentence of abstract (appears similarly elsewhere in text). If possible it might be helpful to have a native English speaker look over the manuscript to make minor changes throughout to aid in readability. At times the authors make use of rather metaphorical language ("empathic podium”, “empathic building”, etc.) which I feel is a bit distracting, but this may be a stylistic choice.

Since data files are included, it would be helpful to label them in such a way as to make them quite easy to navigate if other researchers are interested in exploring them (e.g. including clear descriptive labels and denoting how variables were coded).

Experimental design

Although I think that directly comparing yawning rates across bonobos and humans is certainly interesting, I worry that the differences in methodology between the two species mean that any comparisons can only be highly speculative. Specifically, it seems as though strongly vs. weakly bonded dyads were operationalized differently. Although differences in operationalization are certainly understandable and likely necessary, it makes it difficult to directly compare patterns of yawning. For this reason, I think that the primary contributions of this paper pertain to within species results. For example, it is noteworthy that the authors replicate previous findings in showing that bonobos and humans show more yawn contagion when the trigger yawn comes from a relatively close social partner. Therefore, it might be advisable to place less emphasis on cross species comparisons or at least discuss differences in methodologies between the two species and how these may have influenced results.

If the authors want to retain the current focus of the manuscript, then clarifying their motivation and predictions would be helpful. The authors motivate their study in part by claiming that if humans are the most empathic animal species then they should show the highest empathic response also at the level of emotional contagion. I think that they need to defend this basic premise, especially given that they state that this should always be the case, regardless of relationship quality between the trigger and responder. Undoubtedly humans are able to empathize with others in ways that other animals are not, but it’s not clear that this means they should show the highest level of emotional contagion in this experimental context, and certainly not that they should always do so regardless of relationship quality. As the authors themselves point out, empathy is a multifaceted phenomenon. Humans could be the “most” empathetic not because they experience greater emotional contagion, but rather because they can understand others’ perspective more easily and simulate others’ emotional experiences even in the absence of explicit external cues.

Lines 43: “and therefore…empathetic species”. Perhaps include some sort of reference.

Lines 80-84: The authors seem to imply that monkeys and apes experience Schadenfruede. I’m not sure that this claim is supported. Also, Davila-Ross et al. 2008 appears to be missing from the references.

Line 132: The authors should explain why data was recorded only when at least five subjects were present. Was this also true of the bonobos?

Line 146: How did the authors determine whether a yawn was heard by other group members?

Line 150: Maybe I missed this, but how was the control condition used in analyses?

Line 174-181: Would all bonobo kin dyads have been considered strongly bonded based on the behavior data as well?

Line 220: Please specify why only dyads showing at least three contagion events were included.

Validity of the findings

Comments in the Experimental Design section are also relevant here.

In addition, I think that the authors may need to slightly temper their claims about the relationship between empathy and contagious yawning since it’s not clear how good a measure of empathy contagious yawning actually is (e.g. Bartholomew and Cirulli 2014).

The authors should also avoid making claims about the neural circuitry underling empathic responses unless citations are included. Their own data do not permit them to draw any conclusions about subjective emotional experiences or neural mechanisms.

Line 264: I’m not sure if the authors are claiming that contagious yawning is similar in these species because because both species have fission-fusion social structures. If not consider rephrasing.

Line 288: I would be cautious about attributing a “more developed” emotional and affective sensitivity to bonobos than to chimpanzees. Bonobos have actually been found to exhibit delayed social and cognitive development in some cases (Wobber et al. 2010) which could be interpreted as being “less developed” (although I think interpreting the data in a framework of more vs less developed is problematic generally).

Additional comments

I think that this paper could be of interest and suitable for publication in PeerJ if some of the above issues are addressed.

Reviewer 2 ·

Basic reporting

A generally clear, jargon-free, informative presentation. My suggested revisions are minor.
1) Abstract, line 8, and elsewhere. Another advantage of the naturalistic approach is its avoidance of artifacts associated with differential attention to artificial stimuli such as those provided via video monitors.
2.) Abstract, line 1. Is "echo" a precise metaphor as related to contagion?
3.) Intro, line 1. The critical first sentence needs work, particularly regarding "pinnacles of the human brain." I understand the point being made, but this statement is inaccurate and awkward.
4.) Intro, line 28. Change "on" to "into"
5.) Intro line 33. Perhaps provide other examples of synchronous/contagious behaviors, such as laughing, coughing, and itching/scratching. Provine provides a sample in his book "Curious Behavior: Yawning, Laughing, Hiccupping, and Beyond."
6.) Intro, line 63. Unclear. Probably almost everyone shows some contagion. Perhaps restate as,"Yawn contagion is obvious in human beings--about 50 percent respond to video stimuli of yawning faces (Provine, 1986)--"
7.) Intro, line 66. Perhaps mention that contagion provides a low-tech but significant evidence of mirror-like phenomena.
8.) line 116. Naturalistic approach is good because previous research has demonstrated that awareness of being observed can inhibit yawns.
9.) Why were "Dyads of stranger" excluded?
10.) Discussion, line 256. Break-up the long initial paragraph.
11.) Discussion. General. Regarding the positive correlation between contagion and relationship, mention that prior (and cited) lab studies of human responses to video stimuli always used strangers as stimuli. Thus, the degree of contagion reported in these stimuli of around 50 percept may be a conservative estimate.

Experimental design

OK

Validity of the findings

OK

Additional comments

Interesting, accessible presentation.

---

## Round 0.2 · accepted · Accept

The two reviewers have recommended this revised manuscript for acceptance. One of them, however, did make a few comments that you may want to consider before this manuscript is published.

Reviewer 1 ·

Basic reporting

No comments

Experimental design

No comments

Validity of the findings

No comments

Additional comments

The authors have addressed my major concerns. I now think that the manuscript is appropriate for publication.

Reviewer 3 ·

Basic reporting

See below

Experimental design

See below

Validity of the findings

See below

Additional comments

The ms is much improved and requires only a few tweaks.
1. The subtitle should be changed to read "Emotional affinity matters more than species."
2. Various literary flourishes and flamboyant metaphors are unnecessary, distracting, and may sometimes be inappropriate. Abstract, "awakening and nourishing emotional affinity," "climb down from the empathic podium."
Intro, "emotional and cognitive pinnacles of the human brain" (line 31), "most basal layer of the empathic sphere" (lines 59-60). Are the authors preparing to break out in song?
3. Line 316. Meaning unclear. By "can lack," do the authors mean "is absent"?
4. The authors do not provide an elegant treatment but meet the basic requirements for providing interesting and useful data about contagion and empathy in bonobos.